# In Vitro and In Silico Antimalarial Evaluation of FM-AZ, a New Artemisinin Derivative

**DOI:** 10.3390/medicines9020008

**Published:** 2022-01-24

**Authors:** Ioannis Tsamesidis, Farnoush Mousavizadeh, Chinedu O. Egwu, Dionysia Amanatidou, Antonella Pantaleo, Françoise Benoit-Vical, Karine Reybier, Athanassios Giannis

**Affiliations:** 1UMR 152 Pharma-Dev, Universite de Toulouse III, IRD, UPS, 31400 Toulouse, France; echojay2010@yahoo.com (C.O.E.); karine.reybier-vuattoux@univ-tlse3.fr (K.R.); 2Department of Biomedical Sciences, School of Health, International Hellenic University, 57400 Thessaloniki, Greece; dionusiaam@gmail.com; 3Institute for Organic Chemistry, University of Leipzig, Johannisallee 29, 04301 Leipzig, Germany; seyedehfarnoush.mousavizadeh@uni-leipzig.de; 4Medical Biochemistry, College of Medicine, Alex-Ekwueme Federal University, Ndufu-Alike Ikwo, P.M.B. 1010, Abakaliki 482131, Nigeria; 5Laboratoire de Chimie de Coordination, LCC—CNRS, Universite de Toulouse, 31077 Toulouse, France; francoise.benoit-vical@lcc-toulouse.fr; 6Department of Biomedical Sciences, University of Sassari, 07100 Sassari, Italy; apantaleo@uniss.it

**Keywords:** novel artemisinin derivatives, in silico study, artemisinin resistance, ROS, LC-MS

## Abstract

Artemisinin-based Combination Therapies (ACTs) are currently the frontline treatment against *Plasmodium* *falciparum* malaria, but parasite resistance to artemisinin (ART) and its derivatives, core components of ACTs, is spreading in the Mekong countries. In this study, we report the synthesis of several novel artemisinin derivatives and evaluate their in vitro and in silico capacity to counteract *Plasmodium falciparum* artemisinin resistance. Furthermore, recognizing that the malaria parasite devotes considerable resources to minimizing the oxidative stress that it creates during its rapid consumption of hemoglobin and the release of heme, we sought to explore whether further augmentation of this oxidative toxicity might constitute an important addition to artemisinins. The present report demonstrates, in vitro, that FM-AZ, a newly synthesized artemisinin derivative, has a lower IC_50_ than artemisinin in *P. falciparum* and a rapid action in killing the parasites. The docking studies for important parasite protein targets, PfATP6 and PfHDP, complemented the in vitro results, explaining the superior IC_50_ values of FM-AZ in comparison with ART obtained for the ART-resistant strain. However, cross-resistance between FM-AZ and artemisinins was evidenced in vitro.

## 1. Introduction

In 2015, Prof Youyou Tu was awarded the Nobel prize for physiology or medicine for her discovery of artemisinin (ART), a sesquiterpene lactone isolated from *Artemisia annua* [1,2]. In fact, ART is a fast acting drug that targets blood-stage *P. falciparum*, and its derivatives (ARTs) are the most useful antimalarial drugs to combat malaria [3,4]. In the early1990s, Lee et al. [5] described that the metabolism of ARTs into dihydroartemisinin appeared to be crucial for the production of the more polar metabolites, which alkylate heme to form ART-heme adducts [6]. More so, the homolytic cleavage of the endoperoxide bridge [7,8,9] is essential for their antimalarial activity via heme interaction leading to heme alkylation [10] and the alkylation of several biomolecules, such as proteins, lipids [11], and nucleic acids [12]. Soon after the report on the structure and activity of artemisinin, researchers had synthesized a plethora of artemisinin analogues and tested their pharmacological properties [13,14,15]. Moreover, in the past, several semi-synthetic ART derivatives were prepared to combat malaria, following different synthetic strategies [16,17,18,19]. It is well accepted that malaria parasites experience a special challenge with oxidative stress when they invade human red blood cells (RBCs) and attempt to use the amino acids of hemoglobin for the proliferation of their progeny [20]. As hemoglobin is consumed, heme is released. Because heme can catalyze a Fenton type reaction, the parasite mustbecome free of the heme without releasing free iron (Fe^2+^). The solution evolved by the parasite has been to polymerize released heme into a polymer, termed hemozoin, which is largely inactive in catalyzing reactive oxygen species (ROS) production [21,22]. The parasite also produces glucose-6-phosphate dehydrogenase (G6PDH) to facilitate an extra production of reduced glutathione (GSH) [23] in an effort to control the oxidative stress. Importantly, when the heme polymerization capability is compromised (i.e., by the action of many antimalarial drugs such as chloroquine, piperaquine, mefloquine, etc.), the oxidative stress probably becomes too prominent for the parasite to manage and the parasite dies [24,25]. That is why a reduced import of hemoglobin and its breakdown to yield heme (ART activator) leads to reduced the heme-based activation of ART, a decrease of ROS production, and consequently ART resistance [26,27,28]. 

An antimalarial that will remain effective even after a reduced import of hemoglobin and its degradation to heme will offer hope in the fight against ART resistance. Taking into consideration the action of the parasites in the host erythrocyte, novel ARTs were synthesized focusing on ROS generation as a prominent tool to counteract their mitochondrial functions. The 1,2,4-trioxane ring is the key pharmacophore fragment of artemisinin. The construction of this fragment, the most difficult stage both in synthetic and biological approaches of artemisinin production, is carried out during the final synthetic steps with domino one-pot peroxidation/cyclization reactions. Different ART preparations aim to reduce the long-term in vitro ART pressure on initially sensitive parasites and arrest the rise of ART resistance in Southeast Asia [29,30,31]. The intra-erythrocytic oxidative stress induced by ARTs should be investigated to better understand its role in the emergence of artemisinin resistance. Considering the emergence of ART resistance in Southeast Asia and the quest for new artemisinin-based compounds that could substitute the compounds in use, we synthesized new artemisinin derivatives to evaluate their in vitro antimalarial activity in an ART-resistant strain. Moreover, we evaluated the ROS production and the interactions between parasite proteins and its ligands in the presence of the newly synthesized ARTs in order to understand their biology at the molecular level by docking analysis. We examined the *P. falciparum* protein ATP-ase 6 (PfATP6) in detail, due to the fact that ARTs inhibit specifically and selectively the SERCA orthologue (PfATP6) of *P. falciparum* [32]. More so, the major parasite protein involved in the conversion of heme to hemozoin, the heme detoxification protein (PfHDP), was investigated as a possible target of the newly synthesized ARTs [22].

## 2. Materials and Methods

### 2.1. Synthesis of Artemisinin Derivatives 

Our strategy was based on the fact that artesunate, arteether, and artemether are metabolized into dihydroartemisinin, which is the ultimate bioactive derivative [5]. In order to avoid such a process, we decided to synthesize and evaluate the efficacy, selectivity, and oxidative activity of novel artemisinin derivatives containing a onecarbon unit at Position 10 of artemisinin, formally replacing the carbonyl moiety of the parent natural compound (Table 1). The synthesis of these derivatives has been published elsewhere [33].

### 2.2. Cell Cultures 

For the in vitro testing of the efficacy of the molecules, the *P. falciparum* artemisinin-resistant strain F32-ART and its twin artemisinin-sensitive line F32-TEM, selected by Witkowski et al. [34], were used. The parasites were in RPMI 1640 with 5% serum and 2% hematocrit at 37 °C in a humidified 5% CO_2_ atmosphere [35]. For each experiment, the parasites were tightly synchronized by 5% D-sorbitol treatment at the ring stage (0–24 h) [36]. 

Vero cells, a non-cancerous mammalian cell line from the kidney of an African green monkey (*Cercopithecus aethiops*), were used for a safety assessment and subsequent determination of the selectivity indices of the molecules. The Vero cells were cultured in MEM (Dutscher, France) supplemented with 10% fetal bovine serum (Dutscher), 1X non-essential amino acids (Dutscher) 100 U/mL, 100 µg/mL penicillin/streptomycin (Dutscher), and 2 mM L-glutamine (Dutscher) at 37 °C in a humidified 5% CO_2_ atmosphere [37].

### 2.3. Antiplasmodial Activity and Toxicity Investigations

The inhibitory concentration (IC_50_) of the molecules were determined by the SYBR Green method [38]. Tightly synchronized rings at 1% parasitemia were treated with five concentrations (0.1, 1, 10, 100, and 1000 nM) of each molecule, using artemisinin as control, and incubated for 48 h. Each concentration tested was done in triplicate. The molecules were then washed off three times in PBS. The parasites were lysed by freezing/thawing and subsequently probed with SYBR Green for fluorescence reading. IC_50_ values were determined using GraphPad Prism by drawing the curve: % inhibition vs. log 10 of drug concentration and using a four-parameter dose response curve with Equation (1):(1)$$\%\;inhibition=100-100\;\times \;\frac{Signal\;\left(molecule\right)-Signal\;\left(background\;noise\right)}{Signal\;\left(DMSO\right)-Signal\;\left(background\;noise\right)}$$

To differentiate the efficacy of the molecules against F32-ART and F32-TEM, anddetermine if there is cross-resistance between artemisinins and the molecules to be evaluated, a recrudescence assay was carried out as described by Witkowski et al. [34]. A tight synchrony of each strain at a 3% ring-stage was subjected to high doses of FM-AZ (1 μM) and artemisinin (18 μM) as a positive control. These molecule concentrations were chosen after preliminary experiments to obtain the differential concentration for each strain. The molecules were washed off after 48 h, and the parasites from both strains were placed back in drug-free media. The kinetics were monitored for 30 days for a recrudescence to an initial parasitemia of 3%.

To determine the toxicity of the FM-AZ, an MTT (tetrazolium dye) test was done on the Vero cells. The Vero cells at 10^5^ cells/mL were first incubated for 24 h to enable them to adhere to the walls of the plate before being exposed to five different concentrations (1, 10, 100, 1000, and 10,000 nM for FM-AZ and 17.7, 177, 1770, 17,700, and 177,000 nM for ART as a positive control) for 48 h. MTT tetrazolium dissolved in PBS was added to the cells after the medium was removed by flicking the plates. The cells were incubated for 1 h, and the precipitates formed were dissolved by adding DMSO. The formation of purple crystals due to the reduction of MTT by the cell oxidoreductases is a measure of the cell viability. The absorbance was measured at 570 nm with a BioTek µQuant Microplate Spectrophotometer. *IC*_50_ values were determined using GraphPad Prism, in a similar fashion to what was carried out on *P. falciparum* (% inhibition vs. log 10 of drug concentration), and the percentage of growth inhibition was determined as follows:(2)$$\%\;inhibition=100-100\times \frac{Signal\left(molecule\right)}{Signal\;\left(DMSO\right)}$$

The selectivity index was calculated as
(3)$$SI=\frac{IC_{50}\;on\;Vero\;cells}{IC_{50}\;on\;Plasmodium}$$

### 2.4. Oxidation of RBCs

To oxidize the RBCs, they were suspended at a hematocrit of 30% and incubated with 1 mM phenylhydrazine (PHZ) at 37 °C for 4 h, as previously described [39]. Each reaction was terminated by three washes with phosphate buffer saline containing glucose (PBS-glucose). For all protocols described, untreated controls were processed identically.

### 2.5. Liquid Chromatography Mass Spectrometry Analysis

Superoxide radicals and hydrogen peroxide species were analyzed in FM-AZ-treated RBCs and parasitized red blood cells (pRBCs) after 1 h of incubation at 37 °C by Liquid Chromatography coupled with Mass Spectrometry (LC-MS), as previously described [40,41,42]. An Ultimate 3000 UHPLC system consisting of a solvent organizer SRD-3600 with a degasser, a high pressure binary gradient pump HPG-3400RS, a WPS3000TRS thermostated autosampler, an TCC3000SD oven, a DAD3000 UV-Visible detector (ThermoFisher Scientific, Courtaboeuf, France), and an LTQ-Orbitrap XL ETD mass spectrometer (ThermoFisher Scientific, Courtaboeuf, France) was used. The detection of superoxide radicals was performed with a dihydroethidium (DHE) probe (Sigma-Aldrich, St. Quentin Fallavier, France; Cat. n°: 37291) via the detection of 2-OH-E^+^ and the detection of H_2_O_2_ using a coumarin boronic acid (CBA) probe (Sigma-Aldrich, St. Quentin Fallavier, France; Cat. n°: SY3397819310) through the detection of 7-hydroxycoumarin (COH). Electrospray ionization (ESI) was performed in positive and negative mode for superoxide and hydrogen peroxide, respectively. Quantitative analysis was performed using Xcalibur software for integrating the signal obtained with the corresponding extracted mass (*m/z* 330 for 2-OH-E^+^ and *m/z* 161 for COH) chromatograms. In order to confirm the identity of the detected compounds, the mass spectrometer was used in FTMS mode at a resolution of 15,000 for 2-OH-E^+^ and a resolution of 7500 for COH. For 2-OH-E^+^ detection, chromatographic separation was achieved on a Kinetex EVO C18 column, (2.1 × 100 mm, 1.7 μm particle size) (Phenomenex, Le Pecq, France) at a flow rate of 400 μL/min and column temperature set at 50 °C using an aqueous mobile phase containing acetonitrile. For COH detection, chromatographic separation was achieved on a Kinetex C18 column, (2.1 × 100 mm, 1.7 μm particle size) (Phenomenex, Le Pecq, France) at a flow rate of 500 μL/min and column temperature set at 40 °C using an aqueous phase containing formic acid and acetonitrile. 

Unless indicated, all the reagents and chemicals used for the experiments were supplied by Sigma Aldrich, (Saint-Quentin-Fallavier, France).

### 2.6. Docking Analysis

Docking analysis was carried out on the drug discovery platform, MCULE [43]. All docking tests were performed by considering a 40 × 40 × 40 grid and adopting the default grid spacing (0.375 Å), treating the docking binding pocket as rigid and the ligands as flexible; i.e., all rotatable torsions were released. In detail, we used two theoretical proteins models. The first protein we used was PfATP6 (1U5N), i.e., PDB ID 1U5N [9]. The second protein used for the in silico study was PfHDP, created by ITASSER [44], to solve the 3D structure for PfHDP, as long as PfHDP did not have a significant sequence homology with any of the PDB structures deposited in the protein database [22]. The proteins were prepared with MGlTools-1.5.7. [45] and AutoDock4.2 [46,47]. As previously described [48], specific binding pockets (the active site of PfATP6) included Leu263, Phe264, Gln267, Ile977, Ile981, Ala985, Asn1039, Leu1040, Ile1041, and Asn1042. For PfHDP, the binding pockets were Arg4, Arg186, Phe5, Tyr6, Tyr7, Tyr130, Tyr178, Asn8, Asn174, Leu9, Leu133, His172, His175, His197, Cys173, Ser176, Ile177, Ile184, Ile185, and Pro187. According to the literature, the following were used for the two tested proteins: a binding center for 1U5N (x = 48.114, y = 5.856, and z = 22.852) and for PfHDP (x = 45.140, y = 17.970, and z = 49.550). All ligands were sketched and prepared on the platform, MCULE. The LigPlot+ tool was used to display 2D diagrams. Inhibition constant (Ki) was calculated from the estimated free energy values of ligand binding (E.F.EB., ΔG) for each ligand. For the calculation, we used the following equation: Ki = exp ((∆G × 1000)/(R × T)), where ∆G is the docking energy, R (gas constant) is 1.98719 cal K^−K^ mol^−o^, and T (temperature) is 298.15 K [49]. 

## 3. Results and Discussion

### 3.1. Efficacy and Selectivity of Artemisinin Derivatives

The results revealed that the two artemisinin derivatives tested here are effective against *P. falciparum* at pharmacologically relevant concentrations (<100 nM), with FM-AZ as the most effective (12 nM). Further cytotoxicity testing showed that FM-AZ is very selective for *Plasmodium* with a selectivity index of >1500 (Table 2). The IC_50_ of these derivatives revealed that the in vitro efficacy of FM-AZ is close to dihydroartemisinin (12 nM vs. ≤10 nM) and to artemisinin (12 nM vs. 40 nM). The interesting in vitro IC_50_ of FM-AZ could be attributed to the presence of an azide moiety that potentiates their ROS generating ability [50]). More so, these new derivatives [33] are metabolized into intermediates that are different from those of other ARTs, usually dihydroartemisinin [51]. However, a recrudescence assay on artemisinin-resistant (F32-ART) and artemisinin-sensitive (F32-TEM) strains showed that FM-AZ is more effective against the artemisinin-sensitive strain than on the resistant one, with a recrudescence delay between both strains of more than 8 days. These data indicate that there is a strong cross-resistance between artemisinin and FM-AZ (Table 3) (Appendix A). Therefore, although FM-AZ showed a slightly lower IC_50_ than artemisinin, the reported cross-resistance may indicate that it shares some pathways of pharmacological activity with artemisinins [52], which may be connected to ROS generation. The risk of cross-resistance between newly synthetized artemisinin derivatives and artemisinins has already been evidenced [52,53]. To add more information on the effect of FM-AZ in *P. falciparum,* we analyzed the morphology of the parasites after Giemsa-based staining. Figure 1 shows that the parasite damages were observable after 24 h of incubation in pyknotic parasites treated with concentrations higher than 5 nM. 

### 3.2. Evaluation of ROS Activity after Treatment with PHZ and Artemisinin Derivatives

It has been previously demonstrated that artemisinin activation is connected with a high production of ROS [25] and can be drastically enhanced by the pre-oxidation of erythrocytes by phenylhydrazine [39]. For this reason, the measurement of ROS using liquid chromatography coupled with mass spectrometry (LC-MS) was employed in pre-oxidized RBCs treated with FM-AZ, FM-ES, and ART as a positive control. Pre-oxidized RBCs contain a high level of iron (III) that can be reduced to activate artemisinins [39]. The method is based on the detection of a specific adduct formed into the cell after reaction with DHE and CBA probes for O_2_^−^ and H_2_O_2_, respectively. The determination of superoxide and hydrogen peroxide are based on the detection of their reaction product 2-OH-E^+^ and COH, respectively. As was expected, the newly synthesized molecule FM-ES presented high ROS production similar to artemisinin (Figure 2). Interestingly, oxidized erythrocytes pretreated with FM-AZ showed a significantly higher production of ROS, probably explaining its superior antiplasmodial activity in comparison with FM-ES and ART. Moreover, there is a positive correlation between the IC_50_ values of the new antimalarial compounds and the related increase in ROS amounts. 

### 3.3. Molecular Docking Calculation 

Molecular docking calculations were employed to evaluate the ability of FM-AZ, FM-ES, and ART to bind the parasite proteins, PFATP6 (1U5N) and the *P. falciparum* heme detoxification protein (PfHDP), in order to better understand the in vitro activity of these compounds. In detail, the 3D surface structure of the theoretical model proteins PfATP6 and PfHDP is presented in Figure 3, confirming that ART, FM-AZ, and FM-ES interact in the active site of both proteins. The structures of ARTs appear in the same figure.

The free binding energies for the best docking poses of the tested ARTs and the estimated inhibition constant (Ki) were calculated from the estimated free binding energy and are shown in Table 3. The hydrophobic interactions and H-bonds of the ARTs with the tested proteins are presented in Table 4 and Table 5. The illustration of the ligand interactions with the co-involved amino acids in the tested protein targets provides a variety of ligand binding sites and are shown in Figure 4 and Figure 5. The computational approach revealed that the newly synthesized FM-AZ possesses a binding energy very similar to that of ART for the PfATP6 and PfHDP proteins (−6.6 kcal/mol and −6.7 kcal/mol vs. −6.8 kcal/mol and −6.6 kcal/mol, respectively). Unexpectedly, FM-ES presented a lower estimated binding energy for the PfATP6 protein in comparison with the other ARTs but not for the PfHDP protein. The estimation of the inhibition constants (Ki) of all tested compounds was achieved for an easier interpretation of the obtained results. In accordance with the free binding energy results, the docking simulation showed that the lowest value in the PfHDP protein was for FM-AZ (10.36 μΜ) and the lowest value in PfATP6 was for FM-ES (10.36 μΜ). Taking into consideration the in vitro results and the superior efficacy of FM-AZ, one could assume the importance of the PfHDP protein due to the superior Ki observed in comparison with the other ARTs. We observe that the stability of the molecules comes with hydrogen bonds and hydrophobic interactions. Indeed, to better differentiate the three tested compounds, FM-AZ and ART presented a hydrogen bond with an amino acid, Ile 1041, and hydrophobic interactions. Specifically, the FM-AZ hydrogen bond was formed with the azide N of the compound and presented the shortest distance (2.94 Å). On the other hand, FM-ES presented only hydrophobic interactions, did not present any hydrogen bond, and may affect the potency of the compound. In addition, the anti-malarial drugs’ connection to the PfHDP protein demonstrated the above information of the action of the compounds. Furthermore, all compounds showed stability through hydrogen bonds and hydrophobic reactions. More specifically, ART showed two hydrogen bonds with the amino acids Tyr134 and Gln139 and hydrophobic reactions. Interestingly, FM-AZ presented three hydrogen bonds with the amino acids Arg4, Tyr130, and His175 and hydrophobic interactions. Nakatani et al. [48] demonstrated the importance of His175 in the crystal growth of Hemozoin in *P. falciparum*. Two of the three hydrogen bonds were formed with the azide N of the compound (the most essential modification of FM-AZ in comparison with ART), and one of these bonds was formed with His175, which appears, in the bibliography, to be an important amino acid for the connection of PfHDP with hemoglobin. Finally, FM-ES presented four hydrogen bonds, three of which were with Arg4 and one of which was with Gln139, and hydrophobic interactions.

## 4. Conclusions

In summary, we have developed two promising artemisinin derivatives with potential antimalarial activities. The data reported here indicate that FM-AZ is a potent antimalarial compound against sensitive and resistant strains of *P. falciparum*. Moreover, a positive correlation between the antiplasmodial activity of the new antimalarial compounds and the related increase in ROS amounts in oxidized erythrocytes was observed. Docking analysis in parasite protein targets, PfATP6 and PfHDP, confirmed the importance of these already identified candidate targets. Specifically, an in silico study revealed a hydrogen bond formation in the PfHDP amino acid between His175 (an important protein in the crystal growth of Hemozoin in *P. falciparum*) and the azide.

## Figures and Tables

**Figure 1 medicines-09-00008-f001:**
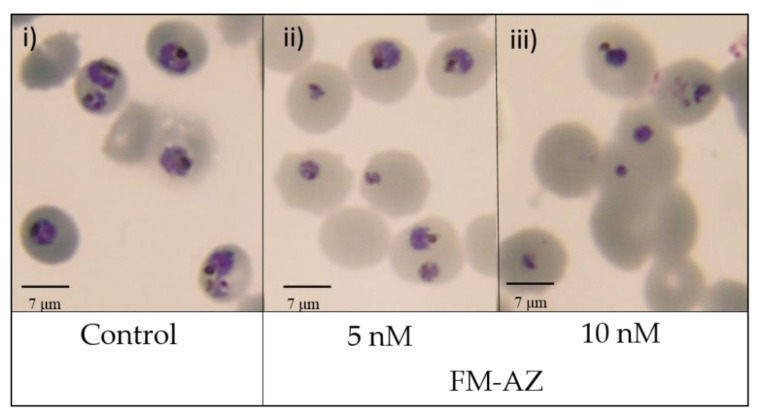
Morphological changes in *P. falciparum* induced by FM-AZ after 24 h of treatment at (**i**) Untreated control (**ii**) 5 nM and (**iii**) 10 nM). Representative images of selected untreated and treated damaged parasites with FM-AZ (5 nM and 10 nM) selected from Diff-Quik^®^ fix-stained thin blood films. The micrographs were obtained using a Leica DM IRB microscope equipped with a 100× oil planar apochromatic objective with a 1.32 numeric aperture, a DFC420C camera, and DFC software version 3.3.1 (Leica Microsystems, Wetzlar, Germany). The scale bar in the figure is 7 µm.

**Figure 2 medicines-09-00008-f002:**
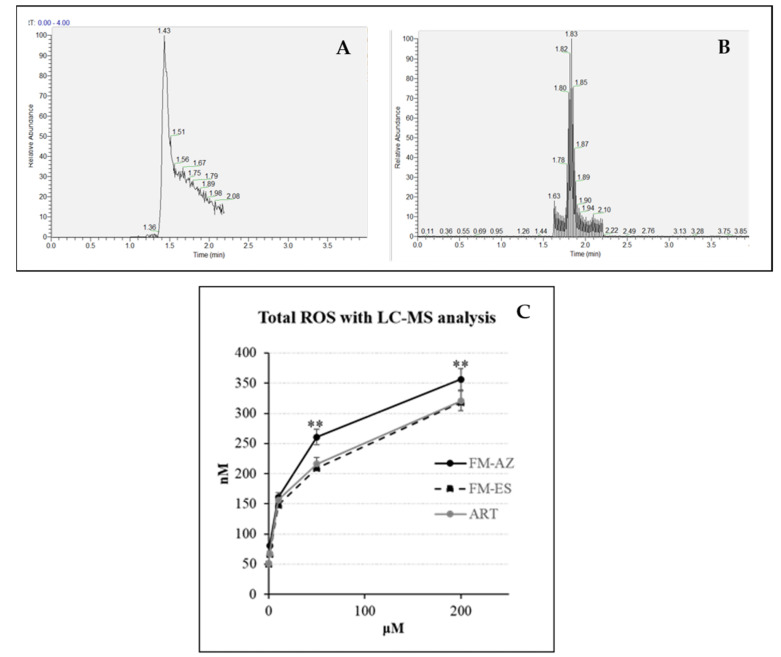
Example of extracted mass chromatograms (**A**) for superoxide radicals and (**B**) hydrogen peroxide species detection, respectively. (**C**) Total amount of ROS (superoxide radicals and hydrogen peroxide) in oxidized erythrocytes after 1 h treatment with 2 new antimalarial compounds (1, 10, 50, and 200 μΜ). ** = *p* value <0.01 between FM-AZ and FM-ES/ART.

**Figure 3 medicines-09-00008-f003:**
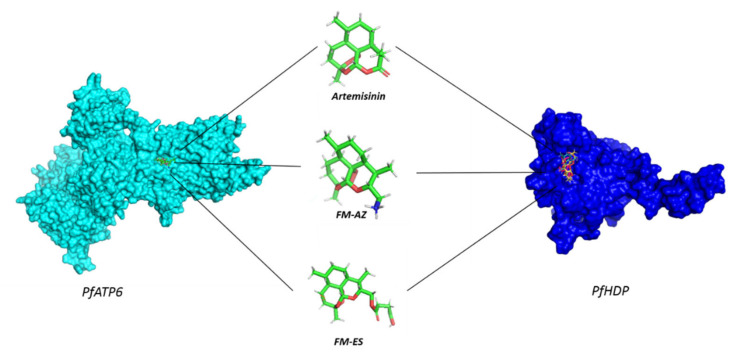
The 3D surface structure of theoretical model proteins, PfATP6 and PfHDP, showing antimalarial drugs interacting in the active site.

**Figure 4 medicines-09-00008-f004:**
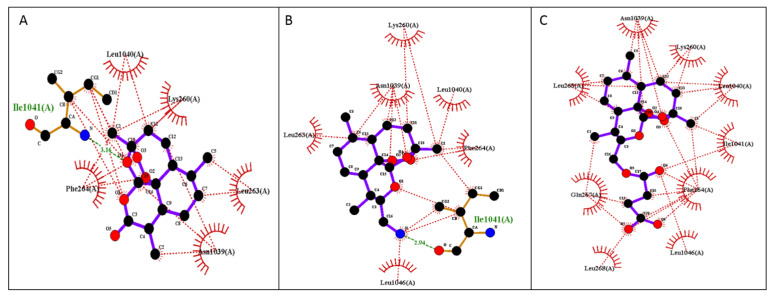
Illustration of 2D plot interactions between amino acids of the theoretical model of PfATP6 and anti-malaria drugs. (**A**) Artemisinin; (**B**) FM-AZ; (**C**) FM-ES.

**Figure 5 medicines-09-00008-f005:**
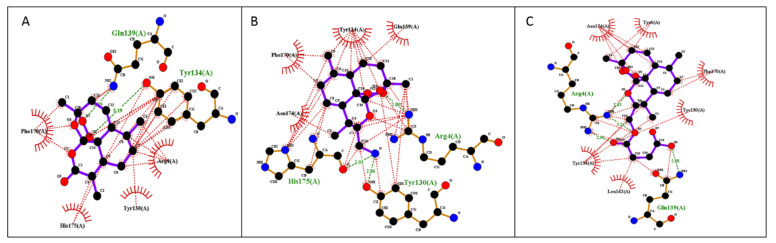
Illustration of 2D plot interactions between amino acids of the theoretical model of PfHDP and anti-malaria drugs. (**A**) Artemisinin; (**B**) FM-AZ; (**C**) FM-ES.

**Table 1 medicines-09-00008-t001:** Molecular weight (g/mol) and chemical structure of artemisinin and the newly synthesized artemisinins (FM-AZ and FM-ES).

	Molecular Weight (g/mol)	Chemical Structure
**FM-AZ**	C_16_H_25_N_3_O_4_ *323.18*	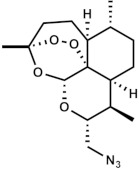
**FM-ES**	C_20_H_30_O_8_ *398.19*	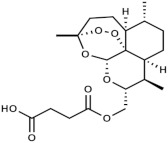
**Artemisinin**	C_15_H_22_O_5_ *282.33*	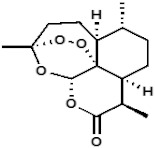

**Table 2 medicines-09-00008-t002:** Potency and selectivity of artemisinin derivatives.

Molecules	IC_50_ (nM)		
	*P. falciparum* F32-TEM strain	Cytotoxicity Vero cells	Selectivity index
FM-AZ	12 ± 7.0	>20.10^3^	>1500
FM-ES	40 ± 0.6	-	-
Artemisinin	40 ± 1.5	160.10^3^ ± 12.10^3^	4000

The experiment was performed only on an ART-sensitive strain because the chemosensitivity assay is not specific enough to differentiate resistant *Plasmodium* from the sensitive ones [54]. Hence, the IC_50_ values of ARTs are usually the same in both the ART-resistant and -sensitive *Plasmodium.* Each datum represents a mean of three independent experiments.

**Table 3 medicines-09-00008-t003:** Recrudescence capacity of *P. falciparum* F32-ART and F32-TEM parasites after 48 h of drug exposure.

Compounds	Doses	Recrudescence Days from 2 Independent Experiments		Delay in Recrudescence Time (Days)
		F32-ART	F32-TEM	
Artemisinin	18 µM	8–8	18–> 30	>10
FM-AZ	1 µM	7–10	15–> 30	>8

Each experiment was performed for F32-ART and F32-TEM cultivated in parallel in the same conditions (adjusted to the same initial parasitemia and cultivated with the same batch of erythrocytes and same batch of human serum) to generate paired results.

**Table 4 medicines-09-00008-t004:** Data showing the binding energy and inhibition constant (Ki) values obtained from the docking analysis of anti-malaria drugs with the theoretical model of proteins PfATP6 and PfHDP.

	PfATP6		PfHDP	
Ligand	E.F.B.E.	Ki	E.F.B.E.	Ki
**Artemisinin**	−6.7	12.27	−6.6	14.53
**FM-AZ**	−6.6	14.53	−6.8	10.36
**FM-ES**	−6.8	10.36	−6.7	12.27

Abbreviations: Estimated free binding energy: E.F.E.B; estimated inhibition constant: Ki. The measurement unit of the estimated free binding energy is kcal/mol and that of the estimated inhibition is μΜ.

**Table 5 medicines-09-00008-t005:** Hydrophobic interactions and H-bonds between anti-malaria drugs and the theoretical model of PfATP6 and PfHDP.

Hydrogens Interactions					
PfATP6			PfHDP		
**Artemisinin**	**FM-AZ**	**FM-ES**	**Artemisinin**	**FM-AZ**	**FM-ES**
Lys260	Lys260	Lys260	Arg4	Arg4	Arg4
Leu263	Leu263	Leu263	Tyr130	Tyr130	Tyr6
Phe264	Phe264	Phe264	Tyr134	Tyr134	Tyr130
Asn1039	Asn1039	Gln267	Gln139	Gln139	Tyr134
Leu1040	Leu1040	Leu268	Phe170	Phe170	Gln139
Ile1041	Ile1041	Asn1039	His175	Asn174	Leu142
	Leu 1046	Leu1040		His175	Phe170
		Ile1041			Asn174
		leu1046			
**H-bond**					
**PfATP6**	**Ligand**	**H-bond**	**Ligand Atom**	**Protein Atom**	**Distance (Å)**
	Artemisinin	1	O_4_	Ile1041:NH	3.16 Å
	FM-AZ	1	N	Ile1041:O	2.94 Å
**PfHDP**	**Ligand**	**H-bond**	**Ligand Atom**	**Protein Atom**	**Distance (Å)**
	Artemisinin	2	O_2_	Tyr134:OH	3.19 Å
			O_3_	Gln139:NE2	2.85 Å
	FM-AZ	3	O_2_	Arg4:NH1	2.80 Å
			N	Tyr130:OH	2.86 Å
			N	His175:O	2.39 Å
	FM-ES	4	O_1_	Arg4:NH1	2.83 Å
			O_5_	Arg4:NH1	3.24 Å
			O_6_	Arg4:NH2	2.98 Å
			O_8_	Gln139:NE2	2.98 Å

## Data Availability

Data is contained within the article.

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
