# Peer review of "In Vitro and In Silico Antimalarial Evaluation of FM-AZ, a New Artemisinin Derivative"

_medicines, 2022, doi:10.3390/medicines9020008_

Round 1

Reviewer 1 Report

Authors of medicines 1514831 manuscript have created two novel artemisinin-like compounds FM-AZ and FM-ES which they evaluated (together with artemisinin as positive control) for anti-plasmodial activity, using two strains (one ART-resistant and the other susceptible) of Plasmodium falciparum. This they did using in vitro approaches: measuring antiplasmodial activity, production of reactive oxygen species (ROS), as well as identification of the ROS using LC/MS. They also did homology modelling and docking of the compounds into the binding sites of two proteins ATPase and a heme detoxification protein. Although the manuscript reports an important findings (as the selectivity and toxicity of the FM-AZ is as good as artemisinin), the manuscript is poorly written. It is very difficult to understand the methods section, likewise the results are poorly written. Too many grammatical errors and typing errors. Manuscript will benefit from addressing all comments raised, and if possible professional editing of English. 

The results also need reorganizing. For example, Table 2 of results (Efficacy and selectivity of artemisinin and new derivatives), authors provided results for the susceptible strain of P. falciparum (F32-TEM), with nothing on the resistant strain. In the methods authors were cryptic. Did not say clearly what strain/s they have used. 

Review – Tsamesidis – et-al – 2021- Medicines-1514831 – MDPI

Abstract

Lines 17-18: Please delete “Therefore, more effective strategies are strongly required.”

Line 18: ‘In this study, we report the synthesis of several novel…’

Line 19. P. falciparum, not Plasmodium falciparum, after the first instance. Here, in line 25 and elsewhere.

Line 20: delete “must” and correct to “devotes”

Line 27: ART in full at the first instance.

Introduction

Line 33: a space between annua and the reference [1,2]. Authors have this issue throughout the manuscript, which has to be corrected. For example, in lines 38, 39, 40, 41, 44, 45, etc.

Line 38: delete interact and keep alkylate.

Line 39: please delete one of ‘the’ repeated.

Line 36 90s not 90s’

Line 39: Is it hemolytic or homolytic?

Line 40: via not in italics.

Line 41: Bridgeford et al., 2019 not formatted. Correct.

Line 44: “…ART derivative were prepare to combat malaria, following different synthetic strategies [16-10]. “

Line 56: Define G6PDH and GSH in full. GSH should be reduced glutathione.

Line 74: Should be ACTs, as this should have been defined in full previously.

Line 77-79: Wrong grammar, correct this sentence.

Line 80: P. falciparum. Same in line 82.

Materials and Methods

Line 90-91: “…in RPMI 1640, with 5% serum, and 2% hematocrit, at 37…”

Line 90: and res of the methods, wherever authors describe important reagents or chemicals provide name of company, city and country in brackets.

Line 103: authors should explain more on how they expose parasite to the concentrations of the molecule. What concentrations exactly?

Line 109 – 110: Font for the formular for % inhibition is different. Same as in under line 127, etc.

Line 113-114: Why 1 micromolar for FM-AZ and up to 18 micromolar for artemisinin?

Line 115: ensure consistency: 48 h not 48 hours.

Line 116: “parasites placed back in drug-free media.

Line 117: what molecule?

Line 119: same here. What is varying concentrations and of what molecule? Important to say what things exactly and give exact concentrations.

Line 129: This should be 2.1: Authors should describe this first before discussion on parasites and exposure methods.

Line 135. Remove brackets from last sentence

Line 141: beginning of this sentence does not make sense. To oxidise what? The whole of this section 2.4 is not informative for non-experts. It needs to be rewritten.

Line 147: Liquid chromatography mass spectrometry analysis.

Line 149: pRBCs in full.

Line 149: Authos keep on using different symbols for degrees Celsius. Please use a single format here and elsewhere.

Line 178: provide a reference for I-TASSER

Line 179: confession?

Line 181-187: An overly long sentence. Must be divided into 3 separate sentence and rephrased to make sense. It’s not understandable at all in present form.

Results and Discussion

Line 196: Who decides <100nM is pharmacologically relevant concentrations?

Line 201: Lower in vitro IC50 of what?

Line 203: What are these different intermediates?

Line 218: Table 2 title should be Efficacy and selectivity of artemisinin derivatives.

Line 218-219: Why is this table only for F32-TEM, what of F32-ART5?

Table 3 F32-ART5 for artemisinin, what does 8-8 mean? Why not 8?

Line 226: biological name in italics.

Line 245: in comparison not in compare.

Line 256-258: Not understandable at all. “ARTs, were reported alone, outside of the pocket of binding and figure 3 presents the 3D surface structure of theoretical model proteins, PfATP6 and PfHDP, confirming that ART, FM-AZ and FM-ES interact in the active site of both proteins.”Please rewrite this.

Lin3 261: hydrophobic not Hydrophobic.

Line 266: -6.6 not -6,6. Please correct all this and elsewhere.

Line 268: in compare again.

Lines 281-282: Not clear at all.

Line 285: end of sentence does not make sense. Hydrophobic interactions?

Line 292: of which. And please correct this sentence as well. It is not clear at all.

Line 297: Table to Table 6: Too many tables in this manuscript. Combined two of these tables into a single table (Table 5).

Conclusion

Line 313: “…we have developed two promising artemisinin derivatives with potential antimalarial activities.”

Line 315: remove experimental and should be ‘against resistant and sensitive strains of P. falciparum”

Line 319: Sentence does not make sense.

Whole of this conclusion need be rephrased.

Reviewer 2 Report

The manuscript medicines-1514831 describes the pharmacological evaluation of new antimalarial endoperoxides. The pharmacological assays are appropriated to assess the aims of the work, especially the employed RSA [reference 47]. The computational study is important, given the need to understand the mechanism of action. However, prior considering publication, authors are encouraged to revise it.

Major essential review:

  1. Page 3, topic [2.3. Synthesis of artemisinin derivatives]. Authors did not publish the synthesis and chemical characterization of FM compounds. Without a peer reviewer of chemistry, there is a potential risk of publishing a pharmacological paper where the compounds were not validated and examined by external experts. There is a potential risk of misleading interpretation. I strongly advice to avoid it.
  2. Authors are encouraged to explain the chemical difference between Artesunate and FM-ES. It seems the same compound.
  3. Again in the topic [2.3. Synthesis of artemisinin derivatives], it is not clear the necessity to avoid the metabolization of FM-compounds into dihydroartemisinin. Actually, there is no need to avoid metabolization. This is not the main limitation of antimalarial endoperoxides (see reference 47). Authors should clarify the difference between metabolism (hepatic) and bioreductive activation of endoperoxides by ferrous heme.
  4. Table 5, authors should say it is potency instead of efficacy. Efficacy is typically in vivo. Moreover, authors should revise the cytotoxicity for VERO cells. If both FM-AZ and Artemisinin were tested in parallel in the same concentration range, and clearly Artemisinin is less cytotoxic for VERO than FM-AZ, there is no explanation to state that regression analysis of CC50 yielded a value that is higher than instead of being 20 microMolar. In addition to, standard deviation of error should be provided for CC50 values.

Round 2

Reviewer 1 Report

Line 297: Table to Table 6: Too many tables in this manuscript. Combined two of these tables into a single table (Table 5).

Authors Response: We prefer to keep the tables separately as both are too long.

I still feel it is better to combine table 5 and 6 - make them into single spacing. 

Line 256-258: Not understandable at all. “ARTs, were reported alone, outside of the pocket of binding and figure 3 presents the 3D surface structure of theoretical model proteins, PfATP6 and PfHDP, confirming that ART, FM-AZ and FM-ES interact in the active site of both proteins.”Please rewrite this.

Line 90: and res of the methods, wherever authors describe important reagents or chemicals provide name of company, city and country in brackets.
